# Compound Control of Trajectory Errors in a Non-Resonant Piezo-Actuated Elliptical Vibration Cutting Device

**DOI:** 10.3390/mi14101961

**Published:** 2023-10-21

**Authors:** Chen Zhang, Zeliang Shu, Yanjie Yuan, Xiaoming Gan, Fuhang Yu

**Affiliations:** 1College of Mechanical and Electrical Engineering, Nanjing University of Aeronautics and Astronautics, Nanjing 210016, China; 2Jiangsu Jingjiang Instrument Transformer Co., Ltd., 206 Xingang Avenue, Jingjiang 214500, China

**Keywords:** piezoelectric stack actuators, elliptical vibration cutting, piezoelectric hysteresis, PID controller

## Abstract

To improve the machining quality of the non-resonant elliptical vibration cutting (EVC) device, a compound control method for trajectory error compensation is proposed in this paper. Firstly, by analyzing the working principle of non-resonant EVC device and considering the elliptical trajectory error caused by piezoelectric hysteresis, a dynamic PI (Prandtl-Ishlinskii) model relating to voltage change rate and acceleration was established to describe the piezoelectric hysteresis characteristics of EVC devices. Then, the parameters of the dynamic PI model were identified by using the particle swarm optimization (PSO) algorithm. Secondly, based on the dynamic PI model, a compound control method has been proposed in which the inverse dynamic PI model is used as the feedforward controller for the dynamic hysteresis compensation, while PID (proportion integration differentiation) feedback is used to improve the control accuracy. Finally, trajectory-tracking experiments have been conducted to verify the feasibility of the proposed compound control method.

## 1. Introduction

Since elliptical vibration cutting (EVC), equipment can improve machining accuracy and can be used to generate the surface micro-textures [1]. It is widely used in the machining of different hard-to-cut materials, such as aerospace, illumination, micro-motion systems, etc. [2,3]. There are some advantages to using EVC technology compared to the conventional machining method, such as reduced cutting force and cutting heat and increased cutting tool life [4]. According to the working mode of the EVC, it can be classified into the resonant and the non-resonant types. Compared with the resonant EVC device working at the specific resonant frequency, the non-resonant EVC device can output a larger displacement at various frequencies and can be easily controlled by an advanced control algorithm to improve its accuracy of output displacement. Flexible hinge mechanisms are generally used in the non-resonant EVC device, which is normally driven by piezoelectric stack actuators (PSAs) to realize displacement transmission and synthesize an elliptical trajectory [5,6,7]. Since the PSA is directly connected to the flexible hinge mechanism, the output displacement of the PSA is equal to the input of the flexible hinge mechanism. However, the PSA has piezoelectric hysteresis nonlinearity characteristics, which significantly affect the accuracy of the generated elliptical trajectory of the EVC device. Therefore, many studies have focused on designing a control algorithm for improving the performance of the EVC device [8]. Hysteretic nonlinearity usually refers to a system with a delayed response, in which changes in the system’s input are reflected in the output within a certain period. This means that the system’s output is not only related to the current input but also affected by the input history of the previous period. 

There are many mathematical models proposed to explain the hysteresis phenomenon and used to fit and express the input and output relationships. The commonly used hysteresis models are the Duhem model, Bouc–Wen model, Preisach model, and PI model [9,10]. In 1897, Duhem proposed the Duhem hysteresis model, which uses a piecewise exponential curve to approximate the hysteresis characteristic. It is an easy method to be used to establish an inverse model. JinHyoung et al. [11] proposed a rate-dependent Duhem model and obtained the auxiliary function of the Duhem model by polynomial approximation. The model shows an excellent fitting accuracy of the output displacement of PSA with frequency variation. The Bouc–Wen model was proposed by the German mathematician R. Boc in 1967 [12]. Although the Bouc–Wen model has a simple structure, few parameters need to be determined and it is easy to implement in the controller. The accuracy of this model is greatly affected by the initial state. Fung et al. [13,14] used an adaptive differential evolution algorithm to identify the Bouc–Wen model. The experimental results show that the controller based on the model can effectively eliminate the effects of hysteresis and improve the motion accuracy of the positioning platform. In 1935, the German physicist Ferenc Preisach [15] first proposed the Preisach model in the hysteresis effect research center based on ferromagnetic materials. Then, some researchers used this model in the hysteresis modeling of piezoelectric materials and achieved good results. Zhou et al. [16] improved the classic Preisach model and identified the model weight function through the fast Fourier transform method, which improved the fast response capability of the system under frequency conversion signals. The PI (Prandtl-Ishlinskii) model was first proposed by Prandtl [17] in 1928 to describe plastic elastic deformation, which is also a phenomenological hysteresis model based on the hysteresis operator. Duhem model and Bouc–Wen model are usually described by differential equations with complex model structures and difficult-to-determine parameters. The Preisach model and the PI model are obtained by weighted superposition of multiple basic hysteresis operators with fewer model parameters and can accurately describe the hysteresis phenomenon. Moreover, compared to the Preisach model, the expression of the PI model is more concise, and there is an analytical inverse model expression.

Based on the hysteresis models mentioned above, various control methods have been designed to eliminate the hysteresis nonlinearity and reduce the hysteresis error of the PSA. An effective control method is voltage feedforward compensation. The inverse model is solved by the established hysteresis model, and the controller is designed based on the inverse model to linearize the voltage input and displacement output. Mohammad et al. [18] established the inverse of the rate-dependent PI model and applied it to the open-loop control of piezoelectric micro-positioning actuators. Galinaitis et al. [19] proposed an improved inverse Preisach model to compensate for the rate-dependent hysteresis nonlinearity in piezoelectric ceramic actuators. Tang et al. [20] used the Bouc–Wen inverse model to reduce the hysteretic nonlinearity of the system. Combined with the single-neuron PID feedback controller, the position-tracking accuracy of the piezoelectric ceramic platform was greatly improved. Fan et al. [21] proposed a radial basis function neural network combined with a rate-dependent PI model, and a disturbance observer was designed for tracking control of PSAs with input frequencies from 1 to 100 Hz. Kang et al. [22] proposed a new fractional normalized Bouc–Wen (FONBW) model. Compared with the classical Bouc–Wen model, the developed FONBW model has a relatively simple mathematical expression and fewer parameters and can characterize the asymmetric and rate-dependent hysteresis behavior of PSAs. Due to external interference, the feedforward control is hard to compensate for the error of the system output. Thus, the feedforward control method makes it difficult to achieve the ideal linearization. Therefore, feedback links are often added on the basis of feedforward control to improve system accuracy.

Kim. et al. [23,24] developed two non-resonant EVC devices in which the vibration amplitudes were used as feedback signals to design a PID control system. These controlled EVC devices have been used to machine micro-grooves, quadrangular pyramids, and other structures. Zhu et al. [25] used two piezoelectric stack actuators arranged in parallel to drive flexure hinge mechanisms to obtain the elliptical vibration trajectory and then designed the controller with a fuzzy PID control method to generate a wedge shape with a frequency of 40 Hz. The micro-pit machining experiment showed that the non-resonant EVC device has stable displacement output performance. Ren et al. [26] proposed a robust output feedback control model based on an uncertainty and disturbance estimator (UDE) without using a state observer for nonlinear single input single output (SISO) systems. The experimental results of piezoelectric nano-positioning show that the model can achieve high precision and high bandwidth trajectory tracking. Cheng et al. [27] established an adaptive Takagi–Sugeno fuzzy model for the input–output relationship of stick-slip type piezoelectric actuators and realized the accurate control of the end effector. Unfortunately, the stability analysis of the system is based on several specific assumptions, but most of the actual systems can not meet these assumptions.

In summary, numerous works have been conducted by researchers relating to controlling the PSA. Each control method has its advantages and drawbacks. The feedforward control has high efficiency and a simple controller structure, making adjustments to the system before deviations, which meets the requirements for PSA controllers. However, the robustness of feedforward control is poor, and it is prone to under-compensation or overcompensation under external disturbances. Feedback control has good stability and high accuracy, and the system has strong anti-interference ability, but the structure is complex, and the calculation amount is large. The compensation of feedback control to the system always occurs after the deviation. Introducing feedback links based on feedforward control and combining the two methods can improve system control performance and motion accuracy. So, to improve the machining quality of the EVC device, a compound control method for trajectory error compensation is proposed in this paper. This paper is organized as follows: in Section 2, the output trajectory characteristics of the non-resonant EVC device are studied, the dynamic hysteresis model of the PSA in each axis is given, and the hysteresis behavior of the PSA is described. Section 3 shows the identification of the parameters of the model through PSO. The piezoelectric hysteresis model is obtained, and the controller is designed according to the established piezoelectric hysteresis model. In Section 4, trajectory tracking experiments have been carried out. Finally, conclusions are provided in Section 5.

## 2. Hysteresis Model of the Piezoelectric Stack Actuator

In this section, the hysteresis characteristics of the elliptical vibration trajectory of the PSA are presented, and the hysteresis model of the PSA is built.

### 2.1. Hysteresis Characteristics Analysis of the EVC Device

In this study, a non-resonant EVC device is used, which consists of two groups of parallel flexible hinges in Y- and Z-directions, as shown in Figure 1 [28]. The size of the used PSAs (Model: PTJ1500505202, Suzhou Pante Company, Suzhou, China) is 5 × 5 × 20 mm. The parameters of the PSAs are listed in Table 1. As shown in Figure 1a, the PSAs are directly connected to the flexible hinge mechanisms in the Y- and Z-directions and are preloaded by bolts (preload torques are 1.4 N∙m and 0.9 N∙m in the Y- and Z-directions). When PSAs are excited by voltages, the flexible hinge mechanisms are driven to move in the corresponding directions. When sinusoidal voltages with a particular phase difference are applied to PSAs, an elliptical vibration trajectory can be generated in the YOZ plane, as shown in Figure 1c. To keep the system stable, the non-resonant device should work below its first resonant frequency. So, the resonant frequencies of the device were identified using a finite element method (FEM) performed in ABAQUS 2019 software. As shown in Figure 1d,e, the first resonant frequencies are 482.6 Hz and 290.7 Hz in the Y- and Z-directions, respectively.

Based on the working principle of the non-resonant EVC device, it can be found that the displacement of the non-resonant EVC device in each direction is directly related to the output displacement of the corresponding PSA, thus relating to the input voltage of the PSA. By using two displacement sensors, the hysteresis curves are obtained with changing input voltage from 0 to 120 V at the different frequencies of 1, 50, and 100 Hz in Y and Z directions, respectively, as shown in Figure 2. From the figure, it can be seen that there is a nonlinear relationship between the output displacement and the input voltage of the PSA in each direction under different frequencies. This nonlinear relationship between voltage and displacement can be described by the piezoelectric hysteresis models [29,30]. Many kinds of hysteresis models have been built and used for designing controllers of PSAs. Among them, the PI hysteresis model based on the operator class can accurately describe the complex hysteresis phenomenon, and there is a wide range of applications in the design of the controller.

### 2.2. Dynamic Hysteresis Model of the PSA

The PI hysteresis model can be assumed as a weighted superposition of play operators with different thresholds. The relationship between the input and output can be described by the play operator as shown in Figure 3, and the mathematical expression is described as [16]:(1)$$\left\{\begin{array}{c} P_{r}\left[u\right]\left(t\right)=max\left\{u\left(t\right)-r,min\left\{u\left(t\right)+r,P_{r}\left[u\right]\left(t-T\right)\right\}\right\}\\ P_{r}\left[u\right]\left(0\right)=max\left\{u\left(0\right)-r,min\left\{u\left(0\right)+r,y_{0}\right\}\right\} \end{array}\right.$$
where $u\left(t\right)$,$\;P_{r}\left[u\right]\left(t\right)$, r, and T represent the operator input, output, threshold and update period and $y_{0}$ is the initial state of the operator, which is generally 0. The play operators with different thresholds are weighted and superimposed to obtain the expression of the PI hysteresis model:(2)$$x\left(t\right)=\sum_{i=1}^{n}\omega_{i}P_{r_{i}}\left[u\right]\left(t\right)$$
where $x\left(t\right)$ is the output displacement of the PI model at time *t*; $\omega_{i}$ and *n* represent the operator weights and the number of operators. The width and slope of the play operator are directly related to the threshold r and weight ω, respectively, as can be seen from the definition of the play operator.

The PI model constructed by Equation (2) is called a static PI model because the output of the displacement is independent of the input voltage *u* (*t*) change rate. However, in the actual test, it can be found that when the input voltage frequency of the PSA changes, the shape of the output hysteresis loop changes, which means that it exhibits rate-dependent characteristics. Therefore, to enable the PI model to describe the output displacements of the PSA at different input voltage frequencies and improve the model’s accuracy, it is necessary to improve the rate correlation of the static PI model and construct a dynamic PI model. Regarding rate correlation improvement, the literature [14] proposed a weighted superposition of the dynamic threshold play operator and the dynamic weight to describe the rate-dependent hysteresis characteristics. So, the dynamic threshold can reflect the variation characteristics of the hysteresis loop, while dynamic weights can reflect the characteristics of the hysteretic output displacement changes.

Figure 4a shows the hysteresis curve of a non-resonant EVC device with a sinusoidal bias voltage and the corresponding variation in $\dot{u}$, $\ddot{u}$ in the stable period. It can be seen that during the stable period of the hysteresis curve, the voltage change rates in the rising and falling segments also change over time. In this study, the PI model is improved by using segmented dynamic weights while keeping the threshold of the play operator unchanged, which can improve the accuracy of the PI model and establish a rate-dependent and acceleration-dependent PI model. According to the different parts of the input voltage, the hysteresis loop is divided into four parts, and different parameters represent the corresponding weights. Due to the good synchronization between the variation characteristics of the hysteresis loop and the first and second derivative values of the voltage, the four-segment division of the hysteresis loop can be represented by the signs of $\dot{u}$,$\ddot{u}$, as shown in Figure 4b.

Based on the above analysis, the segmented dynamic weights can be expressed as: (3)$$\omega_{i}\left(u\left(t\right)\right)=\omega_{i}+\varphi \left(u\left(t\right)\right),\;where\;\varphi \left(u\left(t\right)\right)=\left\{\begin{array}{c} {\alpha_{1}\dot{u}\left(t\right)+\beta }_{1}\ddot{u}\left(t\right)\;\;\;\;\dot{u}\geq 0,\ddot{u}\geq 0\\ \alpha_{2}\dot{u}\left(t\right)+\beta_{2}\ddot{u}\left(t\right)\;\;\;\;\dot{u}\geq 0,\ddot{u}\leq 0\\ {\alpha_{3}\dot{u}\left(t\right)+\beta }_{3}\ddot{u}\left(t\right)\;\;\;\;\dot{u}\leq 0,\ddot{u}\leq 0\\ {\alpha_{4}\dot{u}\left(t\right)+\beta }_{4}\ddot{u}\left(t\right)\;\;\;\;\dot{u}\leq 0,\ddot{u}\geq 0 \end{array}\right.$$
where $\omega_{i}$ is the weight of the static PI model, and the weight function $\varphi \left(u\left(t\right)\right)$ can reflect the characteristics of the hysteresis loop curve of the PSA in the non-resonant EVC device with the changing frequency. Based on Equations (2) and (3), the axial displacement and driving voltage of the non-resonant EVC device can be expressed as:(4)$$x^{d}\left(t\right)=\sum_{i=1}^{n}\omega_{i}\left(u\left(t\right)\right)P_{r_{i}}\left[u\right]\left(t\right)$$

### 2.3. Model Parameter Identification

The dynamic PI model proposed in this paper is based on the original static PI model and obtained through dynamic segmentation of weights. The relevant parameters involve the static model threshold $r_{i}$ and the weight $\omega_{i}$, and dynamic parameters $\alpha_{1}$,$\;\alpha_{2}\;$,$\;\alpha_{3},\alpha_{4},{\;\beta }_{1}$,${\;\beta }_{2},{\;\beta }_{3},and\;\beta_{4}$. The static PI model is identified from hysteresis loop data of 0 to 120 V sinusoidal bias voltage with a frequency of 1 Hz. The parameters of the improved dynamic PI model are identified by the particle swarm optimization (PSO) algorithm [31,32] from hysteresis loop data of 0 to 120 V sinusoidal bias voltage with frequencies of 1 Hz, 10 Hz, 50 Hz, and 100 Hz.

#### 2.3.1. Static PI Model Identification

By defining $d\left(t\right)$ as the output displacement of an axial PSA of the non-resonant EVC device at time *t*, model error, *e*(*t*), can be expressed as the following equation:(5)$$e\left(t\right)=d\left(t\right)-x\left(t\right)$$

By expressing Equation (2) in vector form, Equation (5) can be written as:(6)$$e\left(t\right)=d\left(t\right)-\omega^{T}\cdot P_{r}\left[u\right]\left(t\right)$$
where the threshold vector $\omega^{T}={\left(\omega_{1},\cdots ,\omega_{i},\cdots ,\omega_{n}\right)}^{T}$, the state vector of the play operator at time *t* is $P_{r}\left[u\right]\left(t\right)={\left(P_{r_{1}}\left[u\right]\left(t\right),\cdots ,P_{r_{i}}\left[u\right]\left(t\right),\cdots P_{r_{n}}\left[u\right]\left(t\right)\right)}^{T}$. 

Therefore,
(7)$$e^{2}\left(t\right)=d^{2}\left(t\right)-2\omega^{T}P_{r}\left[u\right]\left(t\right)d\left(t\right)+\omega^{T}\cdot P_{r}\left[u\right]\left(t\right)P_{r}^{T}[u](t)\cdot \omega$$

Due to the sampling signal being discrete data relating to the time, *t*, in the experiment, the accuracy of the model can be measured by the value of the sum variance $\sum_{n=1}^{s}e^{2}\left(kT\right)$, where t=kT,k=1, 2,⋯,s, *T* is the sampling period, *s* is the number of sampling points.

Then, the optimization function can be defined as:(8)$$\begin{array}{c} f\left(\omega \right)=\sum_{n=1}^{s}e^{2}\left(kT\right)=\sum_{n=1}^{s}d^{2}\left(kT\right)+\omega^{T}\left\{\sum_{n=1}^{s}\left[p\left(kT\right)p^{T}\left(kT\right)\right]\right\}\omega -2\omega^{T}\sum_{n=1}^{s}\left[d\left(kT\right)p\left(kT\right)\right] \end{array}$$
where $f\left(\omega \right)$ is a quadratic polynomial relating to the threshold vector, and the optimal solution can be obtained by solving the minimum $f\left(\omega \right)$ by the quadratic programming algorithm. The specific solving process is described as follows:

(a)In theory, the finer the threshold division, the more play operators will be used, then the higher the accuracy of the PI model can be obtained. However, it will increase the complexity of the static PI model. The operators r have a threshold width of 2r beyond the input width of the initial load period. So, there is no need to define any operator beyond the midpoint of the control input range, i.e.,
(9)$$r_{i}\leq 1/2\max\left(u\left(t\right)\right),\;\;i=1,2,\cdots ,n$$

In this paper, the number of operators is set as 10. Then, the threshold $r_{i}$ is selected at equal intervals from 0 to 60.

(b)Using the quadratic programming algorithm (quadgrog function) in MATLAB to find the minimization solution of Equation (8).(c)The identified parameters of static PI models of the non-resonant EVC device in Y- and Z-directions are listed in Table 2 and Table 3, respectively.(d)Figure 5 shows the fitting results of the static PI model in the Y and Z directions of the non-resonant EVC device at a frequency of 1 Hz. The root mean square (RMS) errors of the static PI model corresponding to 1 Hz frequency in the Y and Z directions are 0.0881 μm and 0.1432 μm, respectively.

#### 2.3.2. Parameter Identification of Dynamic PI Model Using PSO Algorithm

The particle swarm optimization (PSO) algorithm is proposed based on the study of the predation behavior of birds. It uses particles with two attributes: “position” and “speed”, to simulate birds in a bird flock, updating the optimal position of each particle and the optimal position of all particles in each iteration. When the specified number of position updates is reached, or the evaluation requirements of the fitness function are met, the optimal position of all particles is considered as the solution to the problem. The implementation process of the PSO algorithm is shown in Figure 6.

In the parameter identification of the dynamic PI model, identified parameters are expressed in vector form as ($\alpha_{1}$,$\;\alpha_{2}$,$\;\alpha_{3},\alpha_{4},{\;\beta }_{1}$,${\;\beta }_{2},{\;\beta }_{3},\beta_{4}$), each particle is regarded as a candidate solution of the parameter vector, and the optimal solution is the optimal global particle of the iteratively updated algorithm. Defining a *D*-dimensional vector space, *N* is the population of particles, and *M* is the maximum number of iterations of the algorithm. Setting the *i*th $\left(i=1,2,\cdots ,N\right)$ particle position as $x_{i}\left(m\right)=\left(x_{i1},x_{i2},\cdots ,x_{iD}\right)$, and the velocity as $v_{i}\left(m\right)=\left(v_{i1},v_{i2},\cdots ,v_{iD}\right)$ before the mth(m=1,2,⋯,M) iteration. In each iteration, through the defined fitness function $f\left(x_{i}\left(m\right)\right)$, selecting the optimal value $p_{i}\left(m\right)$, which is determined by the particle of itself, and the optimal global value $p_{g}\left(m\right),$ which is determined by all particles; then, each particle updates automatically by tracking these two optimal values. Therefore, after the *m*th iteration, the position and velocity of the *i*th particle are expressed as:(10)$$v_{i}\left(m+1\right)=wv_{i}\left(m\right)+c_{1}r_{1}\left(p_{i}\left(m\right)-x_{i}\left(m\right)\right)+c_{2}r_{2}\left(p_{g}\left(m\right)-x_{i}\left(m\right)\right)$$
(11)$$x_{i}\left(m+1\right)=x_{i}\left(m\right)+v_{i}\left(m+1\right)$$

The root mean square error of the measured displacement $d\left(t\right)$ and the model displacement $x^{d}\left(t\right)$ is used as the fitness function:(12)$$f\left(x_{i}\left(m\right)\right)=\sqrt{\frac{1}{S}\sum_{k=1}^{S}{\left[d\left(t\right)-x^{d}\left(t\right)\right]}^{2}}=\sqrt{\frac{1}{S}\sum_{k=1}^{S}{\left[d\left(kT\right)-x^{d}\left(kT\right)\right]}^{2}}$$
where *S* is the total number of data points of the four groups of hysteresis loops of 1, 10, 50, and 100 Hz, and *T* is the sampling period. The working principle of the PSO algorithm used to identify the dynamic PI model parameters is shown in Figure 7.

The dynamic PI model has four unknown parameters, so the particle search space dimension *D* = 8, the number of particles *N* = 100, the maximum number of iterations *M* = 100, and the initial value of each particle is randomly selected between [−1,1], *w* = 0.5, $c_{1}=c_{2}=2$. The Y and Z axes of the non-resonant EVC device were tested separately during the experiments. The working voltage is set as a sinusoidal bias voltage of 0–120 V with frequencies of 1, 10, 50, and 100 Hz. By substituting the voltage value from the experimental data into the dynamic PI model composed of particle position $x_{i}$ in the PSO algorithm, the root mean square error can be calculated by the fitness function according to the model output displacement and the actual displacement. The identified parameters of the dynamic PI model are listed in Table 4 and Table 5.

#### 2.3.3. Simulation Results

The constructed dynamic PI model is used to fit the hysteresis loop curves at different frequencies. Assuming the input voltage $u\left(t\right)$ is a sinusoidal voltage from 0 to 120 V, the output displacement $x^{d}\left(t\right)$ can be obtained by the hysteresis model defined in Equation (4). 

In order to verify the effectiveness of the dynamic PI model, in addition to the static PI model, we also introduced the generalized rate-dependent Prandtl-Ishlinskii (GRPI) model to compare with the dynamic PI model. A GRPI model using a rate-dependent play operator is described as follows:(13)$$\left\{\begin{array}{c} P_{h}\left[u\right]\left(t\right)=max\left\{h_{l}(u,\dot{u})-r,min\left\{h_{r}\left(u,\dot{u}\right)+r,P_{h}\left[u\right]\left(t-T\right)\right\}\right\}\\ P_{h}\left[u\right]\left(0\right)=max\left\{h_{l}\left(u\left(0\right),\dot{u}\left(0\right)\right)-r,min\left\{h_{r}\left(u\left(0\right),\dot{u}\left(0\right)\right)+r,y_{0}\right\}\right\}\\ h_{l}\left(u,\dot{u}\right)=a_{1}u\left(t\right)+a_{2}\left|\dot{u}\left(t\right)\right|\\ h_{r}\left(u,\dot{u}\right)=b_{1}u\left(t\right)+b_{2}\left|\dot{u}\left(t\right)\right|\\ x_{h}\left(t\right)=\sum_{i=1}^{n}\omega_{i}P_{h_{i}}\left[u\right]\left(t\right)\; \end{array}\right.$$
where $u\left(t\right)$,$\;P_{h}\left[u\right]\left(t\right)\;and\;T$ represent the rate-dependent play operator input voltage, output update period, $h_{l}(u,\dot{u})$ and $h_{r}\left(u,\dot{u}\right)$ are the dynamic envelope functions of $u\left(t\right)$ and its derivative $\dot{u}\left(t\right)$, and $a_{1}$,${\;a}_{2}$,${\;b}_{1}$,${\;b}_{2}$ are constants. The threshold of the operator r is same with the classical play operator.

Figure 8, Figure 9, Figure 10 and Figure 11 show the fitting results using the dynamic PI model and GRPI model in Y- and Z-directions of the non-resonant EVC device at two different frequencies (10 Hz and 100 Hz). Table 6 lists the RMS errors of two models under the same input signals. 

From Table 6, it can be observed that the proposed dynamic PI model had the best accuracy compared with the GRPI model and static PI model.

## 3. Controller Design with Dynamic Hysteresis Compensation

The advantage of feedforward control lies in its predictability. However, in practical engineering applications, measuring all disturbances in advance and obtaining accurate predictive models is impossible. At this point, it is necessary to add feedback control, which can correct the deviation of the system in real time under any external interference. Considering the hysteresis characteristics of the output displacement of the PSA used in the non-resonant EVC device and the unpredictable disturbances (temperature, wear, etc.) in the processing process, this paper adopts a compound control method of feedforward and PID control. The inverse dynamic PI model is used to construct the feedforward controller for the feedforward control, while PID feedback is used to reduce the impact of insufficient model accuracy and potential interference and to improve the control accuracy.

The block diagram of the compound control is shown in Figure 12. Where $H^{-1}$ represents the feedforward controller based on the inverse hysteresis model of the non-resonant EVC device. The controller calculates the reference displacement $x_{r}$ to obtain the feedforward voltage $u_{ff}$. The PID feedback controller uses the deviation $e\left(t\right)$ between the actual displacement *x* and the reference displacement $x_{r}\left(t\right)$ to calculate the deviation voltage $u_{fb}$. Then, the $u_{ff}$ and $u_{fb}$ are superimposed to obtain the output voltage signal *u* to apply on the PSA to drive the EVC device. 

The inverse dynamic PI model is used to design the feedforward controller. Because the PI model has an analytical inverse, its inverse model is still a PI model in expression. According to references [33,34,35,36], the inversion formula of the Prandtl-Ishlinskii model is also applicable to the segmented dynamic PI model proposed in this paper. The relationships of thresholds and weights between the dynamic PI model and its inverse model can be obtained directly as:(14)$$\omega_{1}^{-1}\left(x_{r}\left(t\right)\right)=\frac{1}{\omega_{1}\left(x_{r}\left(t\right)\right)}$$
(15)$$\omega_{i}^{-1}\left(x_{r}\left(t\right)\right)=\frac{{-\omega }_{i}\left(x^{d}\left(t\right)\right)}{\sum_{j=1}^{i}\omega_{j}\left(x_{r}\left(t\right)\right)\sum_{j=1}^{i-1}\omega_{j}\left(x_{r}\left(t\right)\right)}i=2,3,\cdots ,n$$
(16)$$r_{i}^{-1}\left(x_{r}\left(t\right)\right)=\sum_{j=1}^{i}\omega_{j}^{-1}\left(x_{r}\left(t\right)\right)(r_{i}-r_{j})i=1,2,\cdots ,n$$
where,$\;x_{r}\left(t\right)$ is the expected displacement, $\omega_{i}^{-1}\left(x_{r}\left(t\right)\right)$ and $r_{i}^{-1}\left(x_{r}\left(t\right)\right)$ are the weight and operator threshold of the inverse PI model, respectively. Then, the inverse dynamic PI model is expressed as follows:(17)$$u_{ff}(t)=u\left[x_{r}\right](t)=\sum_{i=1}^{n}\omega_{i}^{-1}\left(x_{r}\left(t\right)\right)P_{r_{i}^{-1}\left(x_{r}\left(t\right)\right)}\left[x_{r}\right]\left(t\right)$$

Based on the conventional PID control algorithm, the signal $u_{fb}$ is expressed as:(18)$$u_{fb}\left(t\right)=k_{p}\left[e\left(t\right)+\frac{1}{T_{I}}\int_{0}^{t}e\left(t\right)dt+T_{D}\frac{de\left(t\right)}{dt}\right]$$

Since the feedforward control algorithm uses a discrete numerical calculation mode, the PID control algorithm needs to be discretized. After discretization, the PID algorithm in Equation (18) can be expressed as:(19)uk=kpek+TTI∑j=0kej+TDTek−ek−1=kpek+ki∑j=0kejT+kdek−ek−1T
where $k_{p}$ is the scale factor,$\;k_{i}=k_{p}/T_{i}$ is the integral constant, $k_{d}=k_{p}T_{D}$ is the differential constant, T is the sampling period,k is the sampling number.

## 4. Trajectory Tracking of the Non-Resonant EVC Device

The experiments have been conducted to verify the control accuracy of the proposed compound control method on the non-resonant EVC device.

The control program is built into LabView 2018 software. A National Instruments data acquisition card (Model: NI USB–6361X, National Instruments, Austin, TX, USA) was used to acquire data. PSAs made by Suzhou Pante Company (Model: PTJ1500505202, Suzhou Pante Company, Suzhou, China) were used to drive the EVC device. Voltage signals used for PSAs were magnified by a Trek piezo amplifier (Model: PZD350, Advanced Energy Industries Inc., Denver, CO, USA). Vibrations of the non-resonant EVC device in Y- and Z- directions are independently controlled by the same program. Two Micro Sense capacitance sensors (Model: 5300, KLA Company, Milpitas, CA, USA) were orthogonally arranged to measure the tool vibration trajectories. The experimental setup is shown in Figure 13.

The parameters required for the feedforward controller are listed in Table 2, Table 3, Table 4 and Table 5. The parameters $k_{p}$, $k_{i}$, and $k_{d}$ required for the PID feedback control algorithm are obtained by the Cut and Try method. The identified parameters of $k_{p}$, $k_{i}$, and $k_{d}$ in Y-direction are 50, 300, and 0.06, respectively, while the corresponding parameters in Z-direction are 70, 450, and 0.08, respectively.

During the experiments, two different cases were studied based on the ellipse inclination angle. The ellipse inclination angle is defined as the angle between the long axis of the ellipse and the Z-direction. For case 1, the ellipse inclination angle was set as 0° and depicted as a reference track in Figure 14a. The vibration amplitude in the Y-direction was set as within 1–6 μm, while the vibration amplitude in the Z-direction was set within 1–11 μm. For case 2, the ellipse inclination angle was set as 20° and depicted as a reference track in Figure 14b. The corresponding vibration amplitudes in Y- and Z-directions were set within 1–5 μm and 1–9 μm, respectively. For both cases, the vibration frequencies were set as 100 Hz. The measured elliptical trajectories are shown in Figure 14, denoted as actual tracks (orange lines in the electronic version). The measured vibration displacements in Y- and Z-directions are demonstrated in Figure 15 for both cases. In Figure 15, the Y REF and Z REF (solid lines) represent the ideal displacements, while the Y ACTL and Z ACTL (dash lines) represent measured displacements. It can be seen from Figure 15a that the maximum errors of displacements in Y- and Z-directions are 0.308 μm and 0.369 μm, respectively. From Figure 15b, the maximum errors of displacements in Y- and Z-directions are 0.154 μm and 0.252 μm, respectively. For both cases, the relative error is less than 6.2%, which means the proposed compound control method has good control accuracy for this non-resonant EVC device. So, the non-resonant EVC device can output an elliptical vibration trajectory with a higher frequency range with the proposed compound control method.

## 5. Conclusions

To eliminate hysteresis characteristics of the non-resonant EVC device, this paper proposes a compound control method for improving the accuracy of the output displacements of the non-resonant EVC device. The following conclusions can be drawn:A dynamic PI model was built by considering the rate-dependent and acceleration-dependent hysteresis characteristics to improve the accuracy of the PI model. Then, the particle swarm optimization was used to identify the dynamic PI model parameters. Based on the identified parameters, the dynamic PI model was used to fit the hysteresis loop curves of the non-resonant EVC device at different frequencies. The simulation results showed that the proposed dynamic PI model can represent the input rate-dependent and nonlinear properties of the non-resonant EVC device significantly compared with the GRPI and static PI models.Based on the dynamic PI model, its inverse model has been derived. Then, a compound control method has been proposed, in which the inverse dynamic PI model is used as the feedforward controller for the dynamic hysteresis compensation, while PID feedback is used to improve the control accuracy.Finally, based on the proposed compound control method, trajectory-tracking experiments were conducted to verify the feasibility of the proposed compound control method. Experimental results showed that the relative error is less than 6.2%, which means the proposed compound control method has good control accuracy for this non-resonant EVC device.

## Figures and Tables

**Figure 1 micromachines-14-01961-f001:**
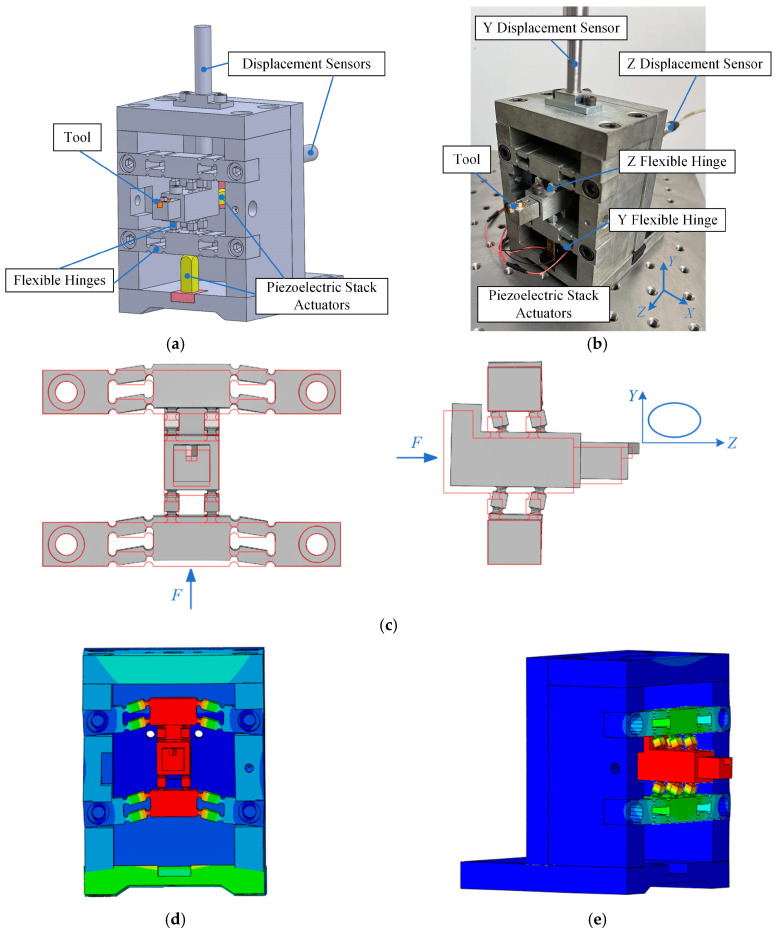
The non-resonant EVC device and its working principle. (**a**) EVC model, (**b**) EVC device, (**c**) working principle of EVC device, (**d**) the first resonant frequency in the Y-direction (482.6 Hz), and (**e**) the first resonant frequency in the Z-direction (290.7 Hz).

**Figure 2 micromachines-14-01961-f002:**
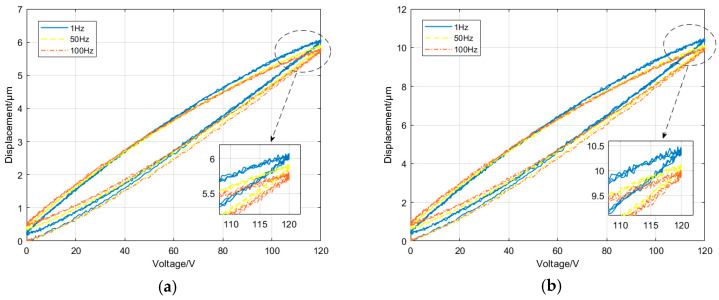
Hysteresis loop curves of the non-resonant EVC device. (**a**) Y-direction and (**b**) Z-direction.

**Figure 3 micromachines-14-01961-f003:**
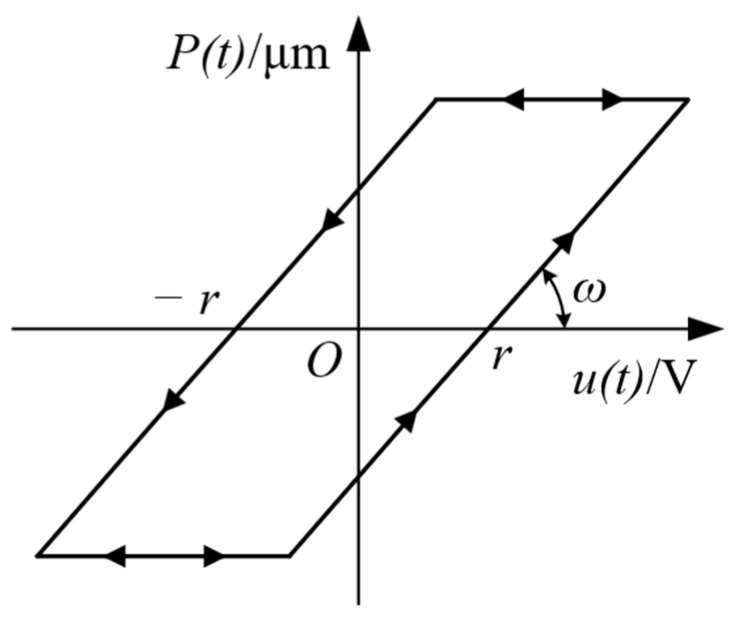
Play operator.

**Figure 4 micromachines-14-01961-f004:**
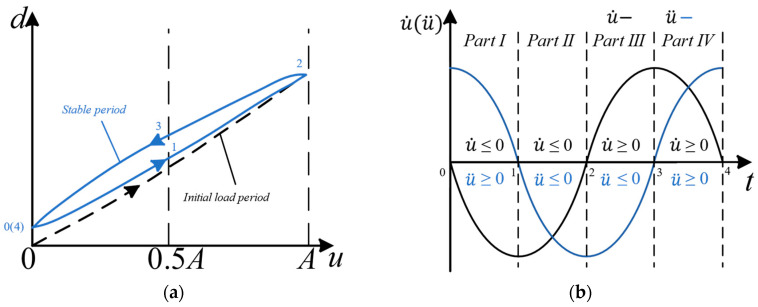
Schematic diagram of hysteresis loop curves. (**a**) the hysteresis curve of EVC device with a sinusoidal bias voltage; (**b**) segmentation of hysteresis curve.

**Figure 5 micromachines-14-01961-f005:**
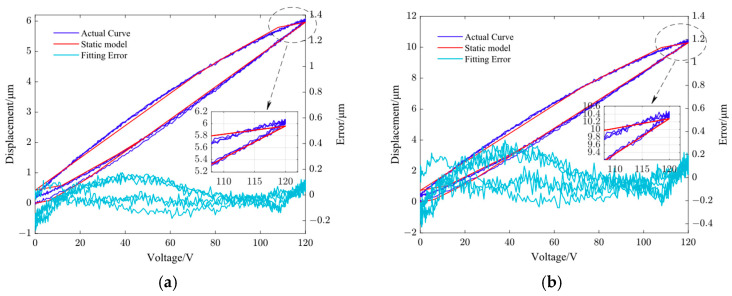
Fitting results of the static PI model of the non-resonant EVC device. (**a**) Y-direction and (**b**) Z-direction.

**Figure 6 micromachines-14-01961-f006:**
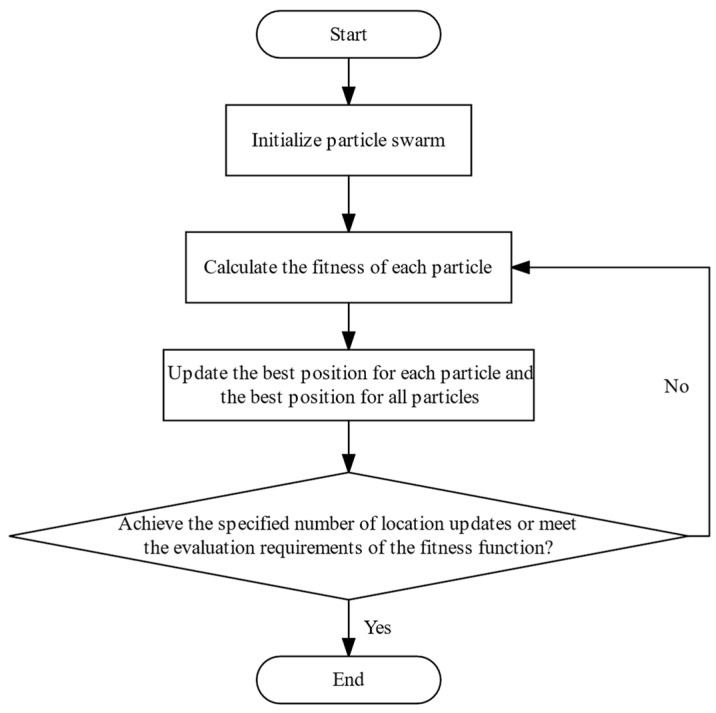
The implementation process of the PSO algorithm.

**Figure 7 micromachines-14-01961-f007:**
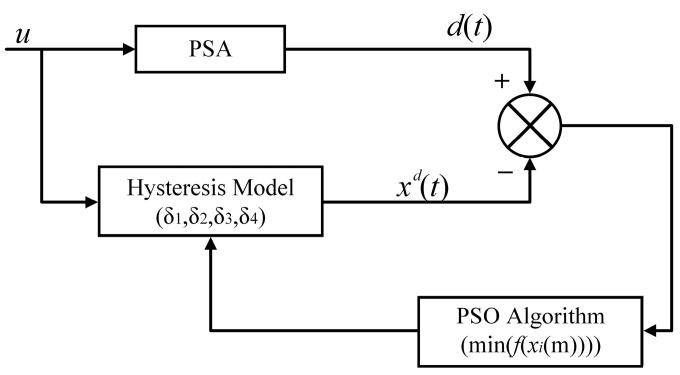
PSO algorithm to identify the dynamical PI model parameters.

**Figure 8 micromachines-14-01961-f008:**
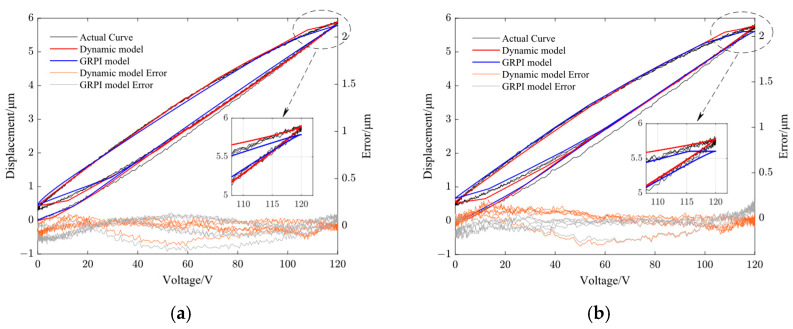
Comparison of dynamic PI model output curve with GRPI model output curve and the model error in Y-direction. (**a**) 10 Hz and (**b**) 100 Hz.

**Figure 9 micromachines-14-01961-f009:**
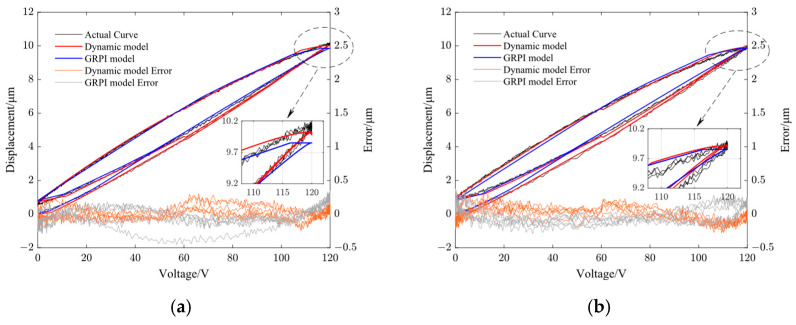
Comparison of dynamic PI model output curve with GRPI model output curve and the model error in Z-direction. (**a**) 10 Hz and (**b**) 100 Hz.

**Figure 10 micromachines-14-01961-f010:**
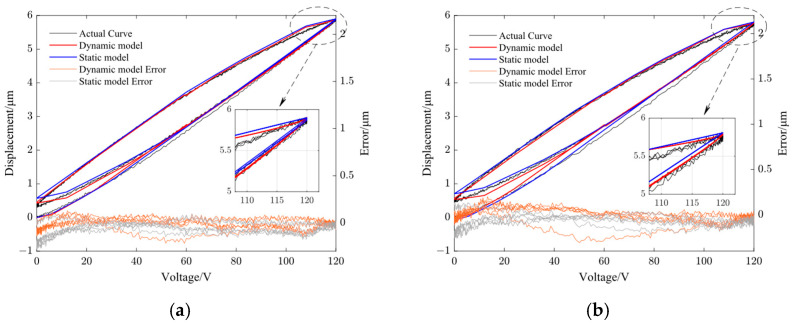
Comparison of dynamic PI model output curve with static PI model output curve and the model error in Y-direction. (**a**) 10 Hz and (**b**) 100 Hz.

**Figure 11 micromachines-14-01961-f011:**
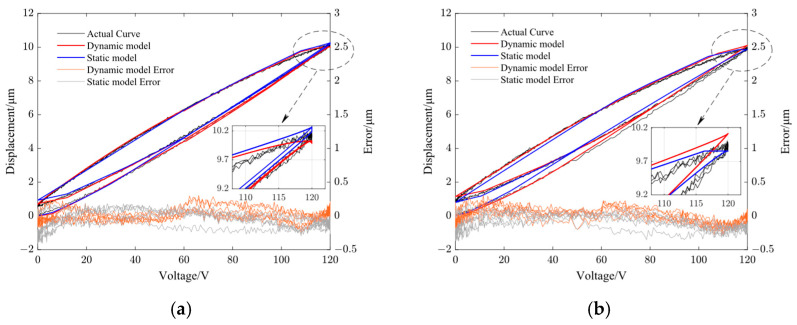
Comparison of dynamic PI model output curve with static PI model output curve and the model error in Z-direction. (**a**) 10 Hz and (**b**) 100 Hz.

**Figure 12 micromachines-14-01961-f012:**
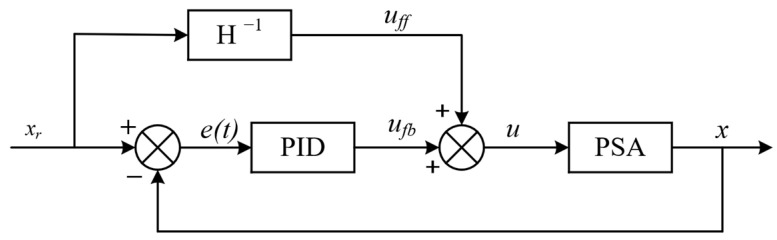
Block diagram of compound control.

**Figure 13 micromachines-14-01961-f013:**
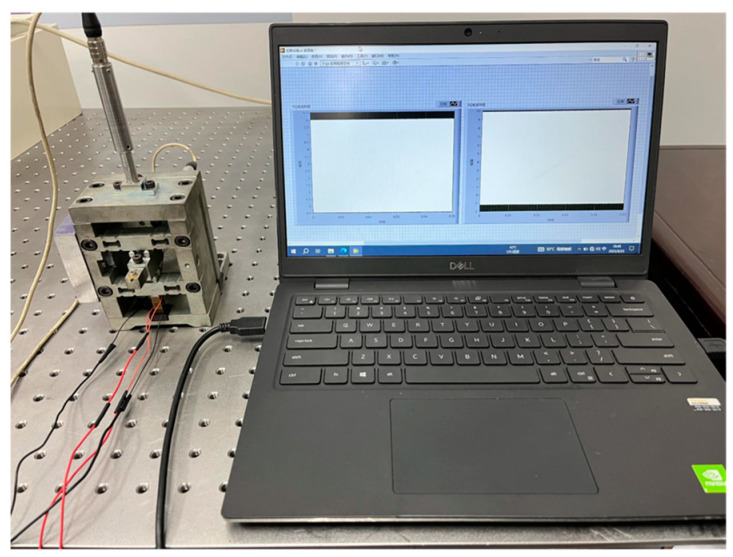
Experimental setup.

**Figure 14 micromachines-14-01961-f014:**
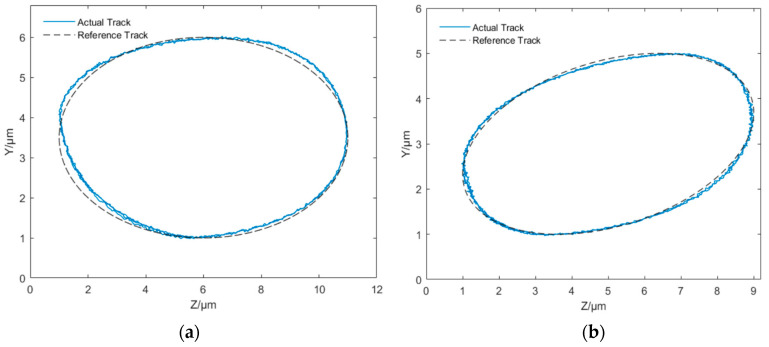
Measured elliptical trajectories of case 1 (**a**) and case 2 (**b**).

**Figure 15 micromachines-14-01961-f015:**
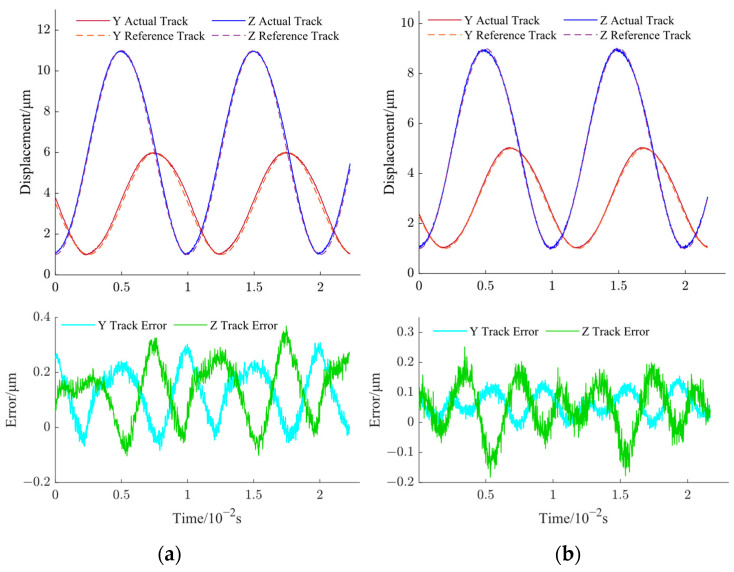
Measured vibration displacements in Y- and Z-directions. (**a**) Case 1 and (**b**) case 2.

**Table 1 micromachines-14-01961-t001:** Parameters of the PSAs.

**Permittivity** $\varepsilon_{r3}^{T}$	Density ρ	$\mathrm{Coupling}\;\mathrm{Factor}\;K_{p}$	$\mathrm{Longitudinal}\;\mathrm{Piezoelectric}\;\mathrm{Strain}\;\mathrm{Coefficient}\;d_{33}$	$\mathrm{Piezoelectric}\;\mathrm{Voltage}\;\mathrm{Constant}\;g_{33}$
3500 ± 20%	7.9 $g/cm^{3}$	70%	650 × 10^−12^ C/N	17 × 10^−3^ $V_{m}/N$
**Elastic compliance constant** $s_{11}^{E}$	**Elastic compliance constant** $s_{33}^{E}$	**Dielectric loss** tanδ	**Quality factor** $Q_{m}$	**Curie temperature** $T_{c}$
14.3 × 10^−12^ m^2^/N	18.5 × 10^−12^ m^2^/N	1.5%	45	240 °C

**Table 2 micromachines-14-01961-t002:** Y-direction static PI model parameters.

*i*	$\mathrm{Thresholds}\;r_{i}$	$\mathrm{Weights}\;\omega_{i}$	*i*	$\mathrm{Thresholds}\;r_{i}$	$\mathrm{Weights}\;\omega_{i}$
1	0	0.01354	6	30	3.3995 × 10^−14^
2	6	0.02790	7	36	6.8298 × 10^−13^
3	12	4.3634 × 10^−8^	8	42	8.9564 × 10^−13^
4	18	0.004014	9	48	8.0574 × 10^−13^
5	24	0.007765	10	54	7.0060 × 10^−15^

**Table 3 micromachines-14-01961-t003:** Z-direction static PI model parameters.

*i*	$\mathrm{Thresholds}\;r_{i}$	$\mathrm{Weights}\;\omega_{i}$	*i*	$\mathrm{Thresholds}\;r_{i}$	$\mathrm{Weights}\;\omega_{i}$
1	0	0.02523	6	30	9.3376 × 10^−11^
2	6	0.04579	7	36	3.3631 × 10^−13^
3	12	1.2251 × 10^−12^	8	42	1.8601 × 10^−14^
4	18	0.004541	9	48	5.8275 × 10^−15^
5	24	0.001636	10	54	3.5902 × 10^−15^

**Table 4 micromachines-14-01961-t004:** Dynamic PI model parameters in Y-direction.

**Parameter**	$\alpha_{1}$	$\alpha_{2}$	$\alpha_{3}$	$\alpha_{4}$
**Data**	−8.4632 × 10^−7^	−3.9336 × 10^−6^	1.4926 × 10^−6^	−3.0164 × 10^−6^
**Parameter**	$\beta_{1}$	$\beta_{2}$	$\beta_{3}$	$\beta_{4}$
**Data**	2.5880 × 10^−7^	9.6875 × 10^−7^	2.1025 × 10^−6^	1.3017 × 10^−6^

**Table 5 micromachines-14-01961-t005:** Dynamic PI model parameters in Z-direction.

**Parameter**	$\alpha_{1}$	$\alpha_{2}$	$\alpha_{3}$	$\alpha_{4}$
**Data**	1.4025 × 10^−5^	9.9453 × 10^−6^	−1.9265 × 10^−5^	−2.8075 × 10^−5^
**Parameter**	$\beta_{1}$	$\beta_{2}$	$\beta_{3}$	$\beta_{4}$
**Data**	1.6927 × 10^−5^	−1.9517 × 10^−5^	−1.5271 × 10^−5^	1.5862 × 10^−5^

**Table 6 micromachines-14-01961-t006:** The RMS error of three models (/μm).

Direction	Frequency	Dynamic PI Model	GRPI Model	Static PI Model
Y	10 Hz	0.0586	0.0933	0.0991
	100 Hz	0.0812	0.0985	0.1075
Z	10 Hz	0.0901	0.1449	0.1406
	100 Hz	0.1039	0.1326	0.1513

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
