# Peer review of "Compound Control of Trajectory Errors in a Non-Resonant Piezo-Actuated Elliptical Vibration Cutting Device"

_micromachines, 2023, doi:10.3390/mi14101961_

Round 1

Reviewer 1 Report

Dear Authors,

My comments and questions are listed in the attachment;

Kind Regards.

Comments are as follows:

Review

The authors of this manuscript present non-resonant elliptical vibration cutting devise driven by two axis piezoelectric stack actuators. Dynamic Prandtl-Ishlinskii model was adopted to establish the piezoelectric hysteresis model for the hysteresis of this device. It is an interesting paper. However, the manuscript should be revised before it can be published.

Main questions or remarks:

1) What are the preload values of piezo stack actuators in the setup? Piezo actuators characteristics change with preload. Please supply this information in the manuscript.

2) Figure 2: Why is the displacement amplitude of Z-direction Piezoelectric stack (10 um) is higher than Y-direction piezoelectric stack (6 um)? Please explain it.

3) Which type of piezoelectric stack actuator have you used in the experiments? Please list some parameters of piezo stack actuators that you use in your setup? For example: What is the value of d33? Please indicate them.

4) What is the frequency response of the setup? Is the first eigenfrequency known? What would be the maximum rate of elliptical excitation can be followed? Is 100 Hz the maximum value? Have you tried higher frequencies? What would be the limitation?

5) Figures 9-12: It seems that RMS error for dynamic PI model is lower compared to the other models. However, absolute value of dynamic errors is at some voltage levels are higher than the one of static model errors seen in these figures. How can you explain this?

6) Have you tried in real application? I mean in the machine tool? Since it would be interesting to see the results when the setup is still precise enough against the cutting forces during machining. What would be the forces exerted by the setup?

Introduction:

Lines 46 - 75: You have explained hysteresis models applied to piezoelectric materials which are mostly quasi-static applications. Since, we know that hysteresis characteristics of piezoelectric materials are rate-dependent of input loads. You have mentioned about that but not adequate. There are few or very limited literatures related to the rate dependency and phenomenological models of piezoelectric materials. I think you should give references about these models as well. For example, the ones below;

[xx] Ang, W. T., Khosla, P. K., & Riviere, C. N. (2007). Feedforward controller with inverse rate-dependent model for piezoelectric actuators in trajectory-tracking applications. IEEE/ASME transactions on mechatronics, 12(2), 134-142.

[xx] Delibas, B.; Arockiarajan, A.; Seemann, W. Rate dependent properties of perovskite type tetragonal piezoelectric materials using micromechanical model. Int. J. Solids Struct. 2006, 43, 697–712.

[xx] Tao, Yi-Dan, Han-Xiong Li, and Li-Min Zhu. "Rate-dependent hysteresis modeling and compensation of piezoelectric actuators using Gaussian process." Sensors and Actuators A: Physical 295 (2019): 357-365.

English should be polished

Reviewer 2 Report

Article Title: Compound control of trajectory errors in a non-resonant piezo-2 actuated elliptical vibration cutting device

The research presented in this article is very interesting and innovative i.e. based on the dynamic PI model, a compound control method has been proposed by authors, in which the inverse dynamic PI model is used as the feedforward controller for the dynamic hysteresis compensation, while PID (Proportion Integration Differentiation) Feedback is used to improve the control accuracy.

The paper is well written and theoretical model was justified with experimental work. The paper can be accepted after minor revision of the following few points mentioned below:

1.     Some more references from latest relevant publications should be added in order to justify the need and uniqueness of the current work.

2.     The statement in the introduction section paragraph 4 line 1-3, “Based on the hysteresis models mentioned above, various control methods have been designed to eliminate the hysteresis nonlinearity and reduce the hysteresis error of the PSA. The most effective control method is voltage feedforward compensation”. Authors have discussed about various control methods, it is suggested that Authors should give reference and comparison of various control methods.

3.     Experimental setup has not shown in the article; authors should provide clear pictorial demonstration of their experimental setup.

Round 2

Reviewer 1 Report

I have no futher questions. 

Moderate editing of English language required